# The Role of Chloroplast Membrane Lipid Metabolism in Plant Environmental Responses

**DOI:** 10.3390/cells10030706

**Published:** 2021-03-23

**Authors:** Ron Cook, Josselin Lupette, Christoph Benning

**Affiliations:** 1MSU-DOE Plant Research Laboratory, Michigan State University, East Lansing, MI 48824-1319, USA; cookron@msu.edu (R.C.); lupettej@msu.edu (J.L.); 2Department of Biochemistry and Molecular Biology, Michigan State University, East Lansing, MI 48824-1319, USA; 3Department of Plant Biology, Michigan State University, East Lansing, MI 48824-1319, USA

**Keywords:** fatty acids, glycerolipids, galactolipids, jasmonate, oxylipin, phospholipids, plant stress

## Abstract

Plants are nonmotile life forms that are constantly exposed to changing environmental conditions during the course of their life cycle. Fluctuations in environmental conditions can be drastic during both day–night and seasonal cycles, as well as in the long term as the climate changes. Plants are naturally adapted to face these environmental challenges, and it has become increasingly apparent that membranes and their lipid composition are an important component of this adaptive response. Plants can remodel their membranes to change the abundance of different lipid classes, and they can release fatty acids that give rise to signaling compounds in response to environmental cues. Chloroplasts harbor the photosynthetic apparatus of plants embedded into one of the most extensive membrane systems found in nature. In part one of this review, we focus on changes in chloroplast membrane lipid class composition in response to environmental changes, and in part two, we will detail chloroplast lipid-derived signals.

## 1. Introduction

The chloroplast is the organelle in plants that synthesizes the bulk of fatty acids (FAs) for the assembly of glycerolipids at the chloroplast envelope membranes and the endoplasmic reticulum (ER). Both compartments extensively interact in the biogenesis of the photosynthetic membranes, necessitating the exchange of lipid precursors across multiple membranes. Phosphatidic acid (PA) is a central metabolite of glycerolipid metabolism, and its synthesis, transport, and conversion are complex and not yet fully understood. In addition, PA affects the activity of multiple enzymes of lipid metabolism, making it often difficult to distinguish its metabolic functions from its signaling functions. As we discuss general aspects of chloroplast lipid metabolism during the first part of this review, a focus is placed on the potential roles of PA and its different molecular species in chloroplast lipid metabolism.

Chloroplast membranes have evolved to accommodate an extensive photosynthetic apparatus while maintaining a minimal dependence on limiting nutrients. While phosphate is a component of most lipids in virtually all other biological membranes, within chloroplasts, it exists in less than half of envelope membrane lipids and less than 15% of thylakoid membrane lipids [1,2,3,4,5]. Instead, these membranes are primarily composed of galactolipids, which are directly derived from photosynthetic products. In addition, sulfolipids are present as an alternative to phosphate-based anionic membrane lipids. This unique membrane system is more than just an economical scaffold for the photosynthetic machinery, it is also mobilized by plant signaling and metabolic networks in response to biotic and abiotic stimuli. In addition to providing background on chloroplast lipid metabolism, the first part of this review describes changes in the lipid classes as a response to stress. During the second part of this review, we discuss the interconnectivity of chloroplast lipid metabolism and lipid-based signaling by oxylipins as one aspect of the dynamic response of chloroplast lipid metabolism to environmental cues.

## 2. Chloroplast Lipid Metabolism

### 2.1. Biosynthesis of Lipid Precursors

Nearly all plant lipid biosynthesis begins with FA biosynthesis in the chloroplast stroma [6], by a Type II FA synthase similar to that of prokaryotes [7] (Figure 1). In the plastid pathway of lipid biosynthesis, the acyltransferase ATS1 transfers 18:1 acyl groups from acyl-acyl carrier protein (acyl-ACP) the *sn*-1 position of glycerol 3-phosphate [8,9]. Then, ATS2 transfers an additional acyl group from ACP to the *sn*-2 position, producing PA at the inner leaflet of the chloroplast inner envelope membrane (IEM) [10]. Since ATS2 is specific to 16:0 acyl-ACP, lipids with a 16-carbon moiety at the *sn*-2 position can be identified as originating from plastid-synthesized PA [11]. The plastid pathway for membrane lipid biosynthesis is also referred to as the “prokaryotic” pathway, although its enzyme components actually have eukaryotic origins [12].

FAs destined for the ER are released from ACP in the stroma by IEM-associated thioesterases, exported, and activated by acyl-CoA synthetases associated with the outer envelope membrane (OEM) [13,14]. Acyl-CoAs are used for PA biosynthesis in the ER just as acyl-ACPs are used in the plastid, with one key difference in substrate specificity: the ER acyltransferase that acylates the *sn*-2 position prefers 18-carbon substrates, not 16-carbon substrates [15]. This allows for lipids with 18-carbon chains at the *sn*-2 position to be identified as derivatives of ER-synthesized PA, or the “eukaryotic” pathway in plants.

### 2.2. Chloroplast Galactolipids

The two primary constituents of chloroplast membranes are monogalactosyldiacylglycerol (MGDG) and digalactosyldiacylglycerol (DGDG) [2,5]. In some plants, including Arabidopsis, tomato, tobacco, and spinach, the *sn*-2 position of MGDG may contain either 16:3 or 18:3 acyl moieties, meaning that both plastid- and ER-assembled PA is directed toward MGDG biosynthesis. Such plants are referred to as 16:3 plants. In contrast, 18:3 plants, which include legumes and monocots, only have 18:3 acyl groups at the *sn-2* position of MGDG, indicating that MGDG is exclusively derived from ER-synthesized PA [16].

Due to the biochemical research focus on spinach and Arabidopsis, galactolipid metabolism is better understood in 16:3 plants than in 18:3 plants. In 16:3 plants, bulk MGDG synthesis under nutrient replete conditions is observed at the IEM, and it requires diacylglycerol (DAG) and uridine diphosphate-galactose (UDP-galactose) as substrates [17]. This reaction is catalyzed by the monogalactosyldiacylglycerol synthase (MGD1), which is associated with the outer leaflet of the chloroplast IEM in 16:3 plants [18,19,20,21]. These 16:3 plants also have PA phosphatase (PAP) activity primarily associated with the IEM, which presumably provides MGD1 with DAG substrate [17]. ER-derived MGDG is synthesized from precursors imported to the IEM by the TRIGALACTOSYLDIACYLGLYCEROL (TGD) complex, although it is still unclear whether PA is the imported species [22]. On the other hand, pea chloroplasts have a major UDP:DAG galactosyltransferase activity in the OEM, which may explain the preponderance of ER-derived galactolipids in 18:3 plants [23]. These 18:3 plants also have far lower PAP activity in chloroplasts envelopes, which was localized to the IEM [11,24,25]. Therefore, MGDG in 18:3 plants may be synthesized at the OEM from ER-derived DAG, while MGDG biosynthesis in 16:3 plants occurs at the IEM from a mixture of plastid-derived PA and ER-derived DAG or PA.

DGDG biosynthesis by a UDP-galactose:MGDG galactosyltransferase (DGD1) was initially observed in pea chloroplast envelopes [26]. The *dgd1* mutant with decreased amounts of DGDG was subsequently isolated in Arabidopsis, leading to the identification of the *DGD1* gene; the DGD1 enzyme was localized to the OEM and determined to require MGDG and UDP-galactose as substrates, which was likely at the cytosolic side of the membrane [27,28,29,30]. Despite equivalent concentrations of plastid-derived MGDG in the OEM and the IEM, DGDG has very low amounts of 16:3 acyl groups, indicating that DGD1 specifically galactosylates ER-derived MGDG [31]. This could be due to substrate preference or to a low abundance of 16:3 MGDG at the outer leaflet of the OEM. DGD1 also contains an N-terminal domain that has been implicated in lipid transfer between the envelope membranes [31].

In 16:3 plants, MGD1 and DGD1 are the primary catalysts for galactolipid biosynthesis in the absence of environmental stress. However, in response to changing biotic and abiotic factors, other isoforms are synthesized or activated that redirect chloroplast lipid metabolism (see below).

### 2.3. Chloroplast Anionic Lipids

In the chloroplast, the anionic membrane lipids phosphatidylglycerol (PG) and sulfoquinovosyldiacylglycerol (SQDG) are both synthesized at the IEM. PG is the only major phospholipid component of the IEM and thylakoid membranes, and its biosynthesis begins with the activation of plastid-synthesized PA to cytidine diphosphate-diacylglycerol (CDP-DAG) by the CDP-diacylglycerol synthases CDS4 and CDS5 [32,33]. Then, PG phosphate synthase exchanges the activated head group for glycerol 3-phosphate, producing PG phosphate (PGP) [34,35,36], which is subsequently dephosphorylated by PGP phosphatase, generating PG [37]. For SQDG biosynthesis, a UDP-sulfoquinovose precursor is produced from UDP-glucose and sulfite by UDP-sulfoquinovose synthase (SQD1) in the chloroplast stroma [38,39]. Then, SQDG synthase (SQD2) synthesizes SQDG from the UDP-sulfoquinovose and DAG at the IEM [40,41].

Phosphatidylinositol (PI), which is found at low concentrations (2–3%) in chloroplast envelopes and thylakoids [42], has recently been implicated in essential developmental and membrane remodeling processes [43]. VESICLE INCLUDING PROTEIN IN PLASTIDS 1 (VIPP1), which is required for thylakoid membrane formation and important for chloroplast membrane stability under various stresses [44,45,46,47], has been found to require a specific interaction with PI phosphates (PIPs) in order to bind and encapsulate membranes [48]. Likewise, the CHLOROPLAST SEC14-LIKE1 protein CPSFL1 is also necessary for thylakoid biogenesis, and it has been shown to transfer PIP between membranes [49]. Taken together, it appears that chloroplast PI has an important role in directing or regulating the biosynthesis of photosynthetic membranes, the specifics of which have yet to be uncovered.

### 2.4. The Roles of Phosphatidic Acid in Chloroplasts

Although PA is the precursor for all other chloroplast glycerolipids, its low abundance means that the quantification of chloroplast PA is difficult [50]. However, studies on PA–protein interactions and transgenic plants with alterations to PA metabolism do provide some preliminary insights into the role of PA beyond its existence as a lipid precursor.

#### 2.4.1. PA Interactions with Proteins of Lipid Metabolism

Several major proteins involved in chloroplast lipid metabolism are known to specifically bind PA (Figure 1). MGD1 has been shown to require allosteric activation by PA and PG in order to synthesize MGDG from DAG and UDP-galactose [51]. Since DAG is itself an inhibitor of PAP activity [52], the PA activation of MGD1 presumably maintains a consistent proportion in the activities of PAP and MGD1. This balance would prevent an excess accumulation of either PA or DAG in the IEM. Based on these discoveries, PA appears to have a typical role in allosteric activation of a metabolic pathway by the initial precursor. In addition, PA may promote MGDG export to the OEM for subsequent DGDG biosynthesis: The N-terminal extension of DGD1 binds specifically to PA, potentially leading to PA-mediated membrane fusion, thereby facilitating galactolipid transfer between the envelope membranes [31].

PA may also be either a substrate or a regulator in the import of ER lipids to the IEM in 16:3 plants, which is a process that is accomplished by the TGD complex [53]. The subunit TGD2 is anchored in the IEM by its N-terminus, while its C-terminus binds specifically to PA; however, the functional role of this interaction is unclear [54]. In addition, the OEM-localized TDG4 protein involved in the import of ER lipid precursors also specifically binds PA, and its PA binding site is required for activity [55,56,57].

Thylakoid membrane biosynthesis may also be regulated by PA. The protein CPSFL1, which is required for vesicle formation at the IEM and thylakoid membrane biogenesis, has a specific binding site for PA and traffics PIP to membranes enriched in PA [49] (Figure 1).

#### 2.4.2. Effects of Modifying Chloroplast PA Metabolism

To better understand the potential regulatory and metabolic roles of PA, the rerouting of lipid precursors to PA biosynthesis was carried out in 16:3 plants by targeting DAG Kinase (DAGK) to specific plastid compartments. In tobacco, the introduction of a bacterial DAGK fused to the N-terminus of the small subunit of rubisco introduced DAGK activity to the chloroplast stroma-facing membranes, although the exact location was not determined. This resulted in an accumulation of ER-derived PA, and subsequently ER-derived PG, in the chloroplast. These transgenic plants exhibited stunted growth, a substantial reduction in chloroplast lipids relative to ER lipids, and a smaller proportion of plastid-derived lipids within the chloroplast [58]. It remains puzzling as to why redirecting both plastid- and ER-derived DAG into PA synthesis at the stroma-side of the chloroplast envelope would result in a disproportionate decrease of prokaryotic galactolipids.

A similar study in Arabidopsis targeted DAGK to chloroplast membrane leaflets facing the stroma, intermembrane space, or cytosol [59]. Surprisingly, DAGK targeted to stroma-facing membrane leaflets did not result in the phenotype witnessed in tobacco, and plant growth and membrane lipid composition was largely unaffected. Further analysis revealed that the majority of DAGK-derived PA in this case was being degraded by phospholipase A activity, preventing a significant increase in PA accumulation. Therefore, excess PA at the IEM inner leaflet is likely responsible for the phenotypes of tobacco lines in which DAGK is targeted to this membrane. In the same Arabidopsis study [59], it was also discovered that DAGK targeted to the intermembrane space of the chloroplast resulted in an increased rate of PA accumulation and stunted plant growth. Taken together, these results suggest that excess PA in the IEM has a negative impact on the development of 16:3 plants.

### 2.5. Membrane Lipid Metabolism under Phosphate Limitation

Upon exposure to phosphate deprivation, plants must re-allocate existing phosphate pools to maintain sufficient levels of nucleic acids and metabolic activity. To this end, they recruit the lipid metabolic network of the chloroplast, which is minimally dependent on phospholipids (Figure 2).

Within the chloroplast, PG is the most abundant phospholipid, comprising approximately 5–15% of chloroplast lipids [1,2,5]. Unlike the galactolipids that constitute the bulk of chloroplast lipids, PG carries a negative charge on its head group. During phosphate deficiency, stable concentrations of anionic chloroplast lipids are maintained by an upregulation of SQDG biosynthesis, which replaces the majority of PG without the need for phosphate [38,40].

A parallel galactolipid biosynthesis pathway in the chloroplast also serves to substitute phospholipids in extraplastidic membranes with galactolipids under phosphate deprivation. Its existence was determined by studying the *dgd1* null mutant of Arabidopsis, in which DGDG biosynthesis was partially restored under limited phosphate availability [60]. The responsible gene, *DGD2*, was found to be expressed during phosphate-limited growth, with the gene product targeted to the outer leaflet of the OEM and requiring MGDG and UDP-galactose as substrates [30,61]. In Arabidopsis, MGDG supplied to DGD2 is produced at the OEM by the gene products of *MGD2* and *MGD3*, which are induced in leaves during phosphate deprivation and require DAG and UDP-galactose as substrates [62,63].

Phosphate limitation-induced conversion of extraplastidic phospholipids into DAG occurs through two known pathways: one relies on a two-step removal of the head group by phospholipase D (PLD) and PAP activity, and the other on phospholipase C (PLC) activity. During phosphate deprivation, *PLDζ1* and *PLDζ2* expression is increased, and *pldζ* null mutants are less capable of converting phospholipids into galactolipids in root tissues [64,65,66]. Enzymes for the subsequent PA dephosphorylation were identified as PA hydrolases, PAH1 and PAH2, two lipins that are active in the cytosol [67,68]. Null *pah1 pah2* mutants are severely deficient in lipid turnover during phosphate limitation, and in contrast to the *pldζ1 pldζ2* mutants, this was also observed in leaf tissue [67]. Therefore, there may be additional PLDs in leaves that provide PAH1 and PAH2 with phospholipid-derived substrate during phosphate limitation.

PLCs also supply phospholipid-derived DAG for galactolipid biosynthesis during phosphate limitation; in particular, expression of the genes encoding the non-specific PLCs *NPC4* and *NPC5* is increased [69,70]. While NPC4 contributes to the majority of PLC activity in leaves under phosphate deprivation, it is located in the plasma membrane and appears to be of little importance in the conversion of phospholipids to galactolipids [69,70]. Meanwhile, the cytosolic NPC5 is responsible for approximately half of the phospholipid-derived DAG that is directed toward DGDG production [70].

### 2.6. Chloroplast Lipid Metabolism during Freezing

Chloroplast galactolipid biosynthesis during normal growth or phosphate stress is achieved through galactosyl transfer from UDP-galactose to DAG or MGDG. However, isolated chloroplasts typically exhibit a significant amount of galactolipid:galactolipid galactosyltransferase (GGGT) activity, which not only yields DAG and DGDG from two molecules of MGDG but can also successively transfer additional galactose monomers from MGDG to make trigalactosyldiacylglycerol (TGDG) and other oligogalactolipids [71]. GGGT-derived DGDG is also unique in that the 1→6 glycosidic bond between the galactose monomers is in the β-configuration rather than the α-configuration [72]. GGGT is associated with the outer leaflet of the OEM, is more active in isolated chloroplasts than it is in vivo during normal growth, and appears to prefer substrates with 16-carbon acyl groups [73,74].

Following characterization of the Arabidopsis mutant *sensitive to freezing 2* (*sfr2*) [75], the gene *SFR2* was identified and proposed to encode an OEM-targeted β-glycosidase that is essential for preventing freeze-induced damage to chloroplasts [76,77]. Then, it was discovered that *SFR2* encodes the enzyme responsible for GGGT activity at the OEM, which is induced by dehydration stresses that include freezing [78] (Figure 2). In freezing-sensitive tomato plants, SFR2 activity prevents damage from salinity- or drought-induced dehydration stress [79]. Dehydration causes the shrinkage of aqueous compartments, and SFR2 activity prevents chloroplast membrane fusion by effectively removing envelope membrane lipids. As SFR2 successively transfers galactosyl moieties from MGDG to adjacent galactolipids, DAG is mostly converted to triacylglycerol (TAG) and sequestered, while the increased abundance of oligogalactolipids augments the aqueous boundary at the membranes [78].

Although the mechanism of SFR activation is not well understood, it may be dependent on its substrate preference of 16-carbon acyl moieties. This preference, first identified in isolated chloroplasts [74], was confirmed by *sfr2* mutant studies, wherein *sfr2* mutants exposed to freezing stress lacked appreciable DAG and TAG with 16:3 acyl groups [78]. In addition, Arabidopsis mutants deficient in the ER pathway for chloroplast lipid biosynthesis were originally identified due to their accumulation of oligogalactolipids [53,56,72,80], possibly a result of an increased abundance of 16:3 MGDG at the OEM. Therefore, a major point at which SFR2 may be regulated is in access to its preferred substrate. In this model, SFR2 activity would result from “uncontrolled” lipid transfer between envelope membranes and within the OEM, which is caused by dehydration or the artificial stresses imposed by chloroplast isolation. However, because this is not a satisfactory model of activation in 18:3 plants, there are likely other activation mechanisms at play. Early studies on isolated chloroplasts showed an abolition of GGGT activity at pH 8.5, and the activation of GGGT by a pH range of 5.9–7.0 and by Mg^2+^ [74,81]. Subsequent in vivo studies demonstrated that SFR2 can be activated by cytosolic acidification or increased cytosolic [Mg^2+^], both of which also result from freezing [82].

## 3. Connection between Thylakoid Lipid Metabolism and Oxylipin Biosynthesis

As described in Section 2, plants adjust to variations in temperature, water, and nutrient availability by reprocessing lipid classes defined by their respective head groups, primarily at the chloroplast membranes. However, chloroplast membrane lipids also have a central role as substrates for acyl moiety modifications in response to such abiotic changes, as well as to biotic stresses such as pathogens or herbivores.

In response to environmental or developmental cues, acyl groups attached to the membrane glycerolipids will be released as free FAs by plastid lipases (Table 1). Lipases are at the nexus of lipid and oxylipin metabolism. Indeed, polyunsaturated fatty acids (PUFAs) are the precursors of a large set of oxidized compounds, called oxylipins [83]. Figure 3 focuses on alpha-linolenic acid (18:3^Δ9,12,15^, *n*-3, ALA) as precursor of oxylipins, but it is important to note that other FAs such as hexadecatrienoic acid (16:3^Δ7,10,13^, *n*-3, HTA) and linoleic acid (18:2^Δ9,12^, *n*-6, LA) are also important precursors of oxylipins in plants.

Then, enzymes such as lipoxygenase (LOXs) and α-dioxygenases (α-DOX) [84] can oxidize these free FAs. LOXs are named 9-LOX or 13-LOX, depending on the oxygenation site on the hydrocarbon chain. At the α-carbon level of the hydrocarbon chain, the heme-containing protein α-DOX catalyzes the formation of an unstable hydroxyperoxide ((2R)-HPOD), which will be directly converted into a shorter-chain fatty acid (17:3 ^Δ8,11,14^ for example) or an aldehyde (heptadecatrienal) [84,85]. The non-heme proteins 9-LOX and 13-LOX catalyze the oxidation of ALA to hydroperoxides: 9-hydroperoxy-10,12,15-octadecatrienoic acid ((9S)-HPOT) and 13-hydroperoxy-10,12,15-octadecatrienoic acid ((13S)-HPOT), respectively [84]. Six isoforms of LOX (LOX1-6) have been described in Arabidopsis thus far. LOX1 and LOX5 are 9-LOX enzymes and LOX2, LOX3, LOX4, and LOX6 are 13-LOX enzymes [86].

(9S)-HPOT and (13S)-HPOT serve as the precursors of a vast library of oxygenated compounds described in Figure 3. One of the most studied groups from this cascade of reactions comprises Jasmonates. Jasmonates include jasmonic acid (JA) and its derivatives such as (+)-7-*iso*-jasmonoyl-_L_-isoleucine (JA-Ile) or methyl-JA (MeJA). JA is notably known to play a key role in the defense against herbivores and necrotrophic pathogens [87]. Jasmonates also affect a wide variety of plant processes such as growth, photosynthesis, or reproduction. 13-allene oxide synthase (13-AOS) catalyzes the formation of 12,13-epoxylinolenic acid (12,13-EOT) from (13S)-HPOT. 12,13-EOT is directly cyclized by allene oxide cyclase (AOC) to *cis*-(+)-OPDA (oxo-10,15-phytodienoic acid). One and four genes, respectively, encode for 13-AOS and AOC (AOC1-4) in *Arabidopsis thaliana*. The *aos* mutant exhibits severe male sterility that can be reversed by the application of MeJA [88]. An alternative route involving HTA as a FA precursor can also form JA. 13-LOX oxidizes HTA into 11-hydroperoxyhexadecatrienoic acid ((11S)-HPHT). 10,11-epoxy-16:3 is formed from (11S)-HPHT via 13-AOS. Then, 10,11-epoxy-16:3 is cyclized to form 12-dinor-*oxo*-phytodienoic acid (*dn*OPDA). Finally, the cyclopentanone ring of OPDA is reduced by a peroxisome-associated enzyme encoded by 12-oxo-phytodienoic acid reductase (OPR3) in Arabidopsis. Then, the product of this reaction will undergo three cycles of β-oxidation to reduce the carboxy-terminal chain to form JA in the peroxisome [84,86]. Recently, Chini and colleagues have also identified OPR3-independent JA synthesis [89]. In this case, OPDA is converted after three rounds of β-oxidation in the peroxisome to 4,5-didehydro-JA (4,5-ddh-JA) and reduced by OPR2 in the cytosol to JA [89].

A related pathway leads to the formation of so-called death acids (DA). This route was identified in *Zea mays* [90]. In this case, the (9S)-HPOT is converted into 9,10-epoxyoctadecatrienoic acid (9,10-EOT) by the 9-AOS and then directly into 10-*oxo*-11,15-phytodienoic acid (10-OPDA) by the 9-AOC. After several cycles of β-oxidation, 10-OPDA is finally converted into DA. DA can also be produced from LA with a pathway similar to that described in Figure 3 with the notable intermediate 10-*oxo*-11-phytoenoic acid (10-OPEA) [90]. 10-OPDA, 10-OPEA, and DA accumulate locally, in particular during southern corn leaf blight (SCLB), which is a fungal disease of maize caused by *Cochliobolus heterostrophus* [90].

Although jasmonates are a highly studied family of oxylipins, they are not the only oxidation products from (9S)-HPOT and (13S)-HPOT. A second family of compounds is the FA divinyl ethers. Divinyl Ether Synthases (DES) synthesize divinyl ether FAs from hydroperoxides [91,92]. A distinction is made between 13-DES, which catalyzes the formation of etherolenic acid, and 9-DES, which leads to the synthesis of colnelenic acid. In Arabidopsis, no gene encoding this enzyme has been identified [84]. The main divinyl ether fatty acids synthesized are colneleic acid, colnelenic acid, etheroleic acid, and etherolenic acid [84]. These compounds seem to have antifungal and antibacterial activities [93,94].

Another interesting family of compounds is the Green Leaf Volatiles (GLVs). These compounds are also synthesized from hydroperoxides. Their synthesis involves the action of several hydroperoxide lyases (9-HPL or 13-HPL), giving rise to a wide variety of six-carbon molecules such as esters, alcohols, or aldehydes (Figure 3) [95], in particular (2E)-hexenal, (3Z)-hexanol, (3Z)-hexenal, or (3Z)-hexenyl acetate. GLVs are volatile organic compounds (VOCs) released by plants after mechanical wounding [96,97]. These compounds are produced during infection by the bacterium *Pseudomonas syringae* in *Phaeseolus lunatus* or *Nicotiana tabacum* [98,99] and by pathogenic fungi such as *Fusarium* spp. in maize [100] or the necrotrophic fungus *Botrytis cinerea* in Arabidopsis [101]. GLVs are also produced in plants during attacks by herbivores [95]. Finally, GLVs can also be synthesized in response to humidity, heat stress, high light, and ozone exposure [99,102].

The oxidation of ALA gives rise to a plethora of derivatives (Figure 3). From (9S)-HPOT and (13S)-HPOT, 9 or 13-hydroxyoctadecatrienoic acid (9 or 13-HOT) can be produced by reduction by a peroxygenase (POX) [103]. Keto FAs can also be synthesized from 9S or 13S-HPOT to form 9 or 13- fatty acid ketotriene (9- or 13-KOT) by a dehydratase mechanism involving LOXs [104]. Finally, trihydroxy FA is synthesized by the coupled action of two enzymes: epoxy-alcohol synthase (EAS) forming epoxy-hydroxy FA and then an epoxy hydrolase (EAH) forming tryhydroxy FA [105,106]. The synthesized trihydroxy FAs show antifungal properties in rice as well as in taro tubers inoculated with the black rot fungus *Ceratocystis fimbriata* [107,108].

Some oxidized lipid-derived compounds are produced non-enzymatically, such as phytoprostanes. Others, such as arabidopsides and linolipins, are examples of oxidized acyl groups formed on complex lipids not requiring the action of plastid lipases (Figure 3). Phytoprostanes are isoprostanoids, i.e., compounds oxidized by a non-enzymatic pathway [109]. Phytoprostanes are in particular synthesized from ALA following a non-enzymatic peroxidation mechanism induced by radical species (singlet oxygen or reactive oxygen species) [110]. Oxylipins that do not require the presence of lipase for their synthesis are Arabidopsides. These compounds were discovered in Arabidopsis in 2001 [111]. Arabidopsides are oxidation products of MGDG and DGDG glycoglycerolipids and contain at least one residue of OPDA or dinor OPDA (dnOPDA) at the *sn*-1 or *sn*-2 position [112,113]. As of the writing of this review, there are seven Arabidopsides molecules named with letters A to G. Finally, a class of compounds recently discovered is linolipins [114,115,116]. There are four linolipins, named A to D. Linolipins A and B are derived from the oxidation of MGDG, while linolipins C and D are the result of the oxidation of DGDG [114,115,116].

Lipid and oxylipin metabolism are tightly linked, and their interconversion is often triggered by environmental stress. Several studies present interesting clues toward a better understanding of this interplay. One example is a functional study carried out in the Arabidopsis *dgd1* mutant [117]. As mentioned above, DGD1 is a protein located in the OEM of the chloroplast and is involved in the conversion of MGDG to DGDG (Figure 1). The *dgd1* mutant exhibits a strong phenotype, notably with a reduction in inflorescence stems, short petioles, and ruffled leaves [27]. The *dgd1* mutant also exhibits a decrease in photosynthetic activity and an alteration in the morphology of the chloroplast [27]. Concerning lipid metabolism, the *dgd1* mutant shows a 90% reduction in the synthesis of DGDG in comparison to the wild type [117] but also a reduction in the level of MGDG [31]. Lin et al. showed that the production of JA, JA-Ile, 12-OPDA, and Arabidopsides was induced in the *dgd1-1* mutant [117], causing the stunted growth of the mutant. The authors proposed that the production of JA in the *dgd1* mutant could depend on an increase in the MGDG/DGDG ratio.

A second example is derived from a study to assess the possible role of JA in lipid remodeling in response to phosphate deficiency in Arabidopsis [118]. Phosphate starvation induces the biosynthesis of JA and its derivative JA-Ile [119]. Using a transcriptomic approach coupled with a lipidomic comparison between the wild type and the mutant *coi1-16*, the authors proposed that JA plays a role in determining the basal levels of PC, PA, MGDG, and DGDG [118]. In addition to the model plant Arabidopsis, studies have also been carried out in other plants. For example, the study of transgenic lines (RNAi and overexpressing lines) of the β-ketoacyl-CoA synthase (*GhKCS13*), involved in the condensation of fatty acids into C16 and C18 generated in the plastid, in cotton *Gossypium* spp. during cold stress led to a change in the composition of sphingolipids and glycerolipids in leaves [120]. In addition, the authors showed that *GhKCS13* affects the biosynthesis of JA in response to cold stress [120]. Indeed, the expression *LOX2* and *AOC* are upregulated in the RNAi lines and are downregulated in the overexpression lines in comparison to wild type, and the synthesis of JA and JA-Ile is induced at 4 °C [120].

### 3.1. Thylakoid Lipid Specific Lipases and their Role in Biotic and Abiotic Stress

Lipases are ubiquitous enzymes found both in eukaryotic and prokaryotic organisms. Lipases are capable of hydrolyzing a large number of substrates such as phospholipids, galactoglycerolipids, or neutral lipids [121]. Thus, depending on the nature of their substrates, the name of the lipases can be specified. For example, phospholipases are lipolytic enzymes hydrolyzing phospholipids at specific ester bonds. Phospholipases are in particular divided into two groups: acylhydrolases (PLA_1_, PLA_2_, PLB) and phosphodiesterases (PLC and PLD). Phospholipase A_1_ (PLA_1_) releases an FA at the *sn*-1 position while phospholipase A_2_ (PLA_2_) releases an FA at the *sn*-2 position. Phospholipases B (PLB) are able to hydrolyze glycerolipids both at the *sn*-1 and *sn*-2 positions. Therefore, acylhydrolases form a free FA and a lysophospholipid as end products. Phospholipases C (PLC) hydrolyze the ester bond between glycerol and the phosphate group, forming a DAG molecule and a phosphoalcohol. Finally, phospholipases D (PLD) will mainly hydrolyze PC to form PA and choline.

Since they are positioned in a strategic location in the synthesis of oxylipins (Figure 3), thylakoid lipid-specific lipases play an important role in responding to environmental stresses. In the context of climate change, discovering the regulatory mechanisms of lipases will help us better understand and engineer physiological responses of plant organisms to environmental stress.

#### 3.1.1. Temperature Variation

Across the seasons, plants are subjected to temperature variations impacting the lipid composition of the membranes, and therefore their biophysical properties. For example, Ruelland et al. showed that the treatment at 0 °C of suspension cells of Arabidopsis resulted in stimulation of PLC and PLD activity, inducing a rapid production of PA [122]. More recently, by bioinformatic analysis with the Arabidopsis e-FB browser, Wang et al. also predicted that *PLASTID LIPASE2* (*PLIP2)* expression could be increased following cold stress [123]. Likewise, in the green microalgae *Chlamydomonas reinhardtii*, a cold stress (4 to 6 °C) of 72 h induces an increase in the expression of *PLASTID GALACTOGLYCEROLIPID DEGRADATION1* (*PGD1*) after six hours, which is an MGDG lipase [124,125]. Finally, the three lipase-like proteins SENESCENCE-ASSOCIATED GENE101 (SAG101), ENHANCED DISEASE SUSCEPTIBILITY1 (EDS1), and PHYTOALEXIN DEFICIENT4 (PAD4), which interact together to form a ternary complex in order to participate in plant defense signaling, are also involved in freezing tolerance in Arabidopsis [126,127,128].

A study focusing on lipase responses during heat stress showed that 24 putative lipase genes are induced during heat stress and return to a basal normal level of expression during recovery [129]. Among these putative lipase genes responding to heat stress are *PLIP2*, *SDP1*, *SDP1-like*, *PLDα2*, *PLDγ2*, *PLDγ3*, *LPPε1, CXE16, PAH1,* and *PAH2*. Higashi et al. focused their efforts on *HEAT INDUCIBLE LIPASE1* (*HIL1*), which encodes a polypeptide of 854 amino acid residues and was defined as a chloroplast MGDG lipase. It releases an 18:3 free FA from 18:3/16:3-MGDG derived from the prokaryotic pathway under heat stress [129].

#### 3.1.2. Osmotic Stress and Drought

Drought is a multicausal environmental stress (low rainfall, temperature variations, salinity, strong light exposure, or even anthropogenic activities) affecting plants. Thus, drought and osmotic stresses can have consequences on the activity of plant and microalgal lipases. For example, the knock-out of pPLAIIα leads to a 25% reduction of water in the leaves compared to wild type [130]. The pPLAIIα-KO plants also exhibit a greater sensitivity to drought compared to wild type [130]. Recently, 194 GDSL-type esterase/lipases (GELPs) genes were identified and classified into 11 subfamilies (A-K) in the soybean genome [131]. The overexpression of *GmGELP28* in Arabidopsis and soybean enhanced tolerance to drought and salt stress in particular, leading to a higher survival rate, chlorophyll and proline contents, and a decrease in H_2_O_2_ and malondialdehyde (MDA) compared to wild type [131]. Finally, it has also been proposed that the expression of *PLASTID LIPASE3* (*PLIP3*) in Arabidopsis could be induced by osmotic stress [123].

Following ^32^P radiolabeling coupled with 300 mM salt treatment for five minutes in the green microalgae *Chlamydomonas moewusii*, Arisz and Munnik propose that the observed increase in Lyso-PA is due to the action of a PLA_2_ lipase on PA generated by DGK [132]. Furthermore, treatment with 100 mM of salt or with 400 mM of sorbitol in *Chlamydomonas reinhardtii* induced an increase in the expression of *PGD1* after 3 h [125].

#### 3.1.3. Pathogen Defenses

Beyond the abiotic environmental stresses, plants are also subjected during their life cycle to stresses caused by pathogenic organisms such as bacteria or fungi. Bacterial pathogens use a type III secretion system (T3S) to promote the introduction of effectors in the host cell during infection, thus altering plant signaling pathways [133,134]. Recently, several studies have shown that lipases may play a role in the defense against pathogenic organisms.

One example concerns Suppressor of AvrBsT-Elicited Resistance 1 (SOBER1), which is a negative regulator of the hypersensitive response of the AvrBsT acetyltransferase effector produced by the pathogenic bacterium *Xanthomonas campestris*. SOBER1 is a member of the α/β hydrolase superfamily, which was identified and characterized in 2007 [135]. Then, Kirik and Mudgett demonstrated that the *sober1-1* mutant accumulated more PA, while the overexpression of *SOBER1* in the *sober1-1* mutant reduced PA levels and inhibited the hypersensitive response [136]. However, by in vitro lipases assays, the authors showed that SOBER1 was a PLA_2_ preferably using PC as substrate and not PA or lyso-PC [136]. In contrast, a second study seems to indicate that AtSOBER1 is a deacetylase and not a PLA_2_ [137]. Finally, a last study shows that SOBER1 is also a suppressor of *Pseudomonas syringae* acetyltransferase effector HopZ5 [138].

A second example belongs to the family of GDSL lipases/esterases. GDSL lipases have a conserved Gly-Asp-Ser-Leu in the amino-acid sequence at the N-terminus [139]. Using a proteomic analysis of the Arabidopsis thaliana secretome treated with salicylic acid (SA) to identify the proteins involved in the plant pathogen response, Oh et al. identified a secreted lipase called GDSL Lipase 1 (GLIP1) [140]. The *glip1* mutant is hypersensitive to infection by the fungal necrotrophic plant pathogen *Alternaria brassicola* [140], while the overexpression of GLIP1 shows a stronger resistance to the pathogens *A. brassicola*, *Pectobacterium carotovorum* (formerly *Erwinia corotovora*), and *Pseudomonas syringae* pv. tomato DC3000 (*Pst* DC3000) [141], thus highlighting the antifungal and antibacterial properties of GLIP1. *GLIP1* expression is induced by treatment with 1.5 mM ethephon (ethylene releaser), but not by SA or JA, highlighting a specific connection between GLIP1 and ethylene (ET) signaling in systemic resistance mechanisms in Arabidopsis [140,141,142,143]. A study carried out with the GDSL Lipase 2 (GLIP2) paralog in Arabidopsis revealed that the respective gene is mainly expressed in the roots and is induced by SA, JA, and ET [144]. The *glip2* mutant exhibits an increased expression of genes encoding auxin metabolism, increased number of lateral roots, and susceptibility to the pathogen *P. carotovorum* [144]. The roles of GLIP1 and GLIP2 lipases in response to pathogens were also studied in *Capsicum annuum* [145] and *Oryza sativa* [146] models.

#### 3.1.4. Oxylipin Responses

All these environmental stresses affecting lipase activity can also trigger a downstream cascade of defense signaling in plants, in particular by inducing Jasmonate metabolism. This section aims to highlight the different observed cases of connecting lipases and oxylipin metabolism in plants.

One example concerns the Plastid Lipases (PLIPs) [123,147]. Three paralogs have been described in Arabidopsis: PLIP1, PLIP2, and PLIP3 [123,147]. These are all PLA_1_ types, but they use different substrates: PLIP1 and PLIP3 prefer 16:1^Δ3*trans*^-PG, while PLIP2 mainly uses MGDG [123,147]. PLIP1 is located in thylakoid membranes, PLIP2 is ubiquitously located in chloroplast envelope membranes, thylakoids, and stroma, and PLIP3 is only present in chloroplast envelope membranes and thylakoids [123,147]. In addition, PLIP1 and PLIP2/3 are functionally separate. *PLIP1* is mainly expressed in seeds and is involved in the biosynthesis of TAGs in Arabidopsis seeds [147]. Furthermore, Aulakh and Durrett also showed that PLIP1 was critical for seed viability of the *dgat1-1* mutant in Arabidopsis [148]. The *PLIP2* and *PLIP3* overexpressing lines show a strong reduction in size due to the synthesis of the active forms of jasmonates (JA, OPDA, JA-Ile, MeJA, 12OH-JA, and 12OH-JA-Ile) and arabidopsides (A, B, and D) [123]. The authors also showed that the expression of *PLIP2* and *PLIP3* was induced two hours after treatment with 7 µM of abscisic acid (ABA). Finally, the triple mutant *plip1,2,3* exhibits hypersensitivity to ABA, which is expressed in particular by a decrease in the germination rate and a yellowish appearance [123].

Another example of the connection between lipases and jasmonate metabolism concerns the study of chloroplast PLA_1_ Defective in Anther Dehiscent1 (DAD1) [149]. The *dad1* mutant is defective in anther dehiscence, pollen maturation, and flower bud opening. These phenotypes resemble those of mutants involved in jasmonate biosynthesis (*dde1, opr3,* and *fad3,7,8* triple mutant) [150,151,152] or the *coi1* mutant, which is insensitive to JA [153,154]. Ishiguro et al. showed that the phenotype of the *dad1* mutant was rescued by the application of 0.1% (*v*/*v*) LA or 500 µM MeJA [149]. Dongle (DGL), a homologue of DAD1 exhibits low PLA_1_ activity but high galactolipase activity for DGDG [155]. Conversely, DAD1 has much lower galactolipase activity compared to DGL for DGDG [155]. Hyun and colleagues proposed that DGL and DAD1 have redundant functions in the biosynthesis of JA during wounding, but they are active at different times: DGL takes part in the early phases of the synthesis of JA, whereas DAD1 would be involved in the late steps of JA production [155]. However, Ellinger et al. showed that DAD1 and DGL were not essential for the biosynthesis of JA during the first sixty minutes after wounding [156]. This study also showed a different localization of DGL at the lipid droplets and not in the plastid [156]. In addition to DGL, DAD1 has five additional homologs, which have been named DAD1-Like Lipase (DALL1-5) [157]. The expression of *DAD1*, *DGL,* and *DALLs* is induced upon wounding, and the expression of *DAD1* and DGL requires the presence of *CORONATINE INSENSITIVE1* (*COI1*), while this is not the case for the DALL1-3 homologs [157]. DALL4 contributes to the biosynthesis of JA in the first hours after wounding [156].

The Glycerolipase A1 (GLA1), which has been studied in the coyote tobacco model [158], is a close PLA_1_ homolog of DAD1 and DGL in *Nicotiana attenuata,* mainly using PC, MGDG, and TAG as substrate [159]. This study revealed that NaGLA1 allows the release of trienoic FA (16:3/18:3) for the biosynthesis of JA in leaves and roots after injury or herbivorous stimulation. However, the authors also showed that NaGLA1 was not essential for the synthesis of GLVs by the HPL pathway as well as for the developmental control of JA biosynthesis in flowers [158].

## 4. Conclusions and Future Perspectives

While the primary function of chloroplasts is to convert light energy into chemical energy, their large and complex membrane networks are a basis on which plants rely to cope with constant environmental fluctuations. Chloroplast membranes are the result of lipid metabolism, which is susceptible to stress-responsive pathways, allowing plants to maintain membrane integrity both inside and outside of the chloroplast. In addition, chloroplast lipids are themselves a reservoir for substrates of a complex oxylipin metabolism, which mobilizes the signaling components of a robust response to biotic and abiotic challenges. As these stress-responsive pathways are far from being completely elucidated, it is important to maintain a focus on understanding the lipid metabolism that underpins them.

A more complete understanding of basal chloroplast lipid metabolism, including transporters involved in lipid trafficking and their specific substrates, will more clearly define the branch points at which the basal metabolism can diverge to stress-responsive pathways. This should be accompanied by progress in the characterization of regulatory components of chloroplast lipid metabolism, including substrate preference and availability for enzymes, allosteric regulation of enzymes by lipids, protein–protein interactions that affect enzyme localization, activity, or turnover, and transcriptional control of the chloroplast membrane proteome. As these aspects of regulation are increasingly studied in the context of various environmental cues, uncovering the roles that chloroplast membrane lipids play within responses to these cues will lead to a greater understanding of how the dynamism and elasticity of their biochemistry allow plants to survive and reproduce as sessile organisms in a sporadic environment.

## Figures and Tables

**Figure 1 cells-10-00706-f001:**
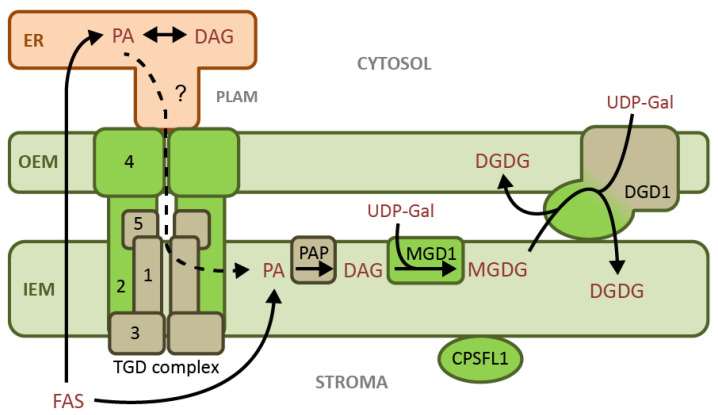
Roles of phosphatidic acid (PA) in chloroplast lipid metabolism. Proteins colored in green have specific interactions with PA, which may serve as a regulator, substrate, or both. The potential role of PA as the substrate for lipid import into the chloroplast is represented by a dotted arrow and a question mark, as this remains uncertain. List of abbreviations in alphabetical order: CPSFL1, CHLOROPLAST SEC14-LIKE1 protein; DAG, diacylglycerol; DGD1, UDP-galactose:MGDG galactosyltransferase; DGDG, digalactosyldiacylglycerol; ER, endoplasmic reticulum; FAS, Fatty acid synthase; MGD1,monogalactosyldiacylglycerol synthase; MGDG, monogalactosyldiacylglycerol; PA, phosphatidic acid; PAP, phosphatidic acid phosphatase; PLAM, plastid associated microsomes; TGD complex, trigalactosyldiacylglycerol complex; UDP-Gal, uridine diphosphate-galactose. The numbers refer to the TGD1-5 proteins forming the TGD complex.

**Figure 2 cells-10-00706-f002:**
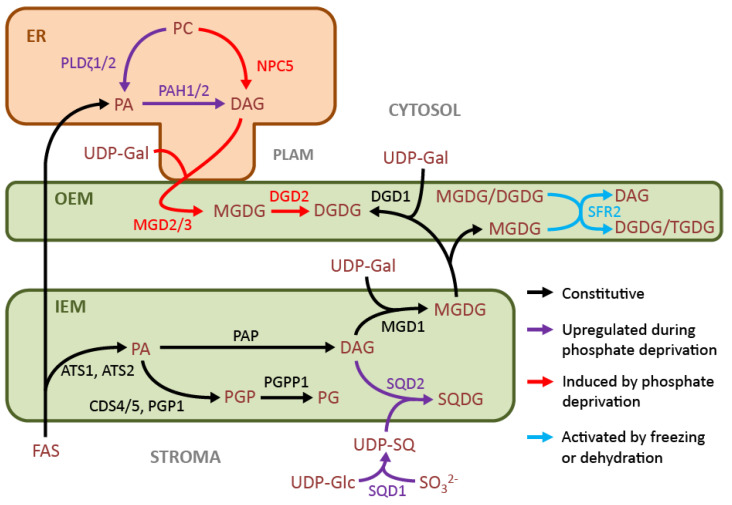
Chloroplast lipid metabolism as a scaffold for metabolic responses to environmental stress. In black, constitutive lipid metabolism in unstressed plants; in purple, constitutive pathways that are upregulated in response to phosphate deprivation; in red, non-constitutive pathways that are turned on during phosphate deprivation; in blue, pathways activated by freezing or dehydration stress. List of abbreviations in alphabetical order: ATS1/2, GLYCEROL-3-PHOSPHATE ACYLTRANSFERASE 1/2; CDS4/5, CYTIDINE DIPHOSPHATE DIACYLGLYCEROL SYNTHASE 4/5; DAG, diacylglycerol; DGD1, UDP-GALACTOSE:MGDG GALACTOSYLTRANSFERASE; DGD2, DIGALACTOSYLDIACYLGLYCEROL SYNTHASE 2; DGDG, digalactosyldiacylglycerol; ER, endoplasmic reticulum; MGDs, monogalactosyldiacylglycerol synthases; MGDG, monogalactosyldiacylglycerol; NPC5, NON-SPECIFIC PHOSPHOLIPASE C5; PA, phosphatidic acid; PAH1 and PAH2, PHOSPHATIDIC ACID PHOSPHOHYDROLASE1 and 2; PAP, PHOSPHATIDIC ACID PHOSPHATASE; PC, phosphatidylcholine; PG, phosphatidylglycerol; PGP, phosphatidylglycerol phosphate; PGPP1, PHOSPHATIDYLGLYCEROPHOSPHATE PHOSPHATASE1; PLAM, plastid associated microsomes; *PLDζ1/2, PHOSPHOLIPASES D ZETA1/2.* SFR2, SENSITIVE TO FREEZING2; SQD1, UDP-sulfoquinovose synthase; SQD2, SQDG synthase; SQDG, sulfoquinovosyldiacylglycerol; TGDG, trigalactosyldiacylglycerol; UDP-Gal, uridine diphosphate-galactose; UDP-Glc, uridine diphosphate-glucose; UDP-SQ, uridine diphosphate-sulfoquinovose.

**Figure 3 cells-10-00706-f003:**
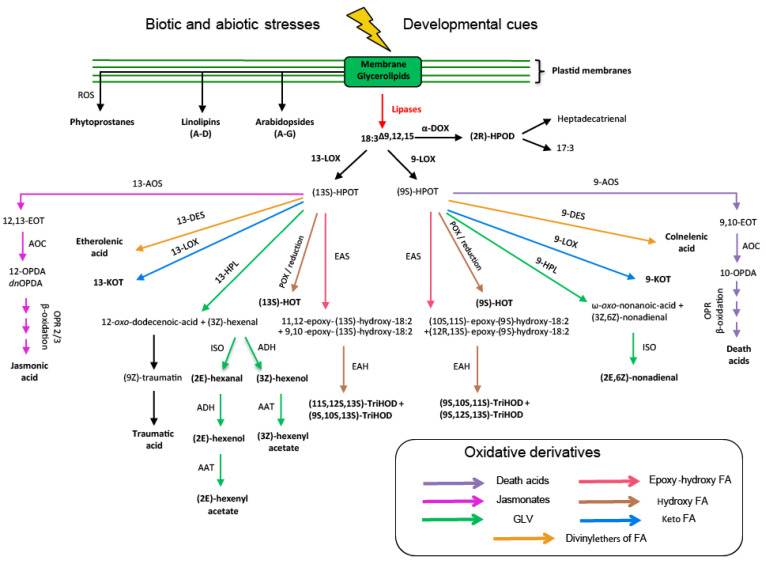
Overview of derivatives oxidized from alpha-linolenic acid. Following environmental stress or developmental cues, alpha-linolenic acid from plastid membrane glycerolipids is released by plastid lipases to be converted into multiple compounds within the metabolism of oxylipins. α-dioxygenases catalyze the formation of unstable hydroperoxide, which are quickly converted to shorter-chain fatty acids or aldehyde, while lipoxygenases catalyze the formation of hydroperoxides, which will serve as precursors for a large set of oxygenate compounds. This library of compounds includes the family of jasmonates (magenta arrows), death acids (purple arrows), green leaf volatiles (green arrows), epoxy–hydroxy fatty acid (pink arrows), hydroxy fatty acid (brown arrows), keto fatty acids (blue arrows) and divinyl ethers of fatty acids (orange arrows). List of abbreviations in alphabetical and numerometric order: (2R)-HPOD, unstable hydroperoxyde; 9-AOS, 9-allene oxide synthase; 9-DES, 9-divinyl ether synthase; 9-HPL, 9-hydroperoxide lyase; 9-LOX, 9-lipoxygenase; 9,10-EOT, 9,10-epoxyoctadecatrienoic acid; (9S)-HPOT, 9-hydroperoxy-10,12,15-octadecatrienoic acid; (9S)-HOT, 9-hydroperoxide lyase; 10-OPDA, 10-*oxo*-11,15-phytodienoic acid; 12,13-EOT, 12,13-epoxylinolenic acid; 12-OPDA, 12-oxo-10,15-phytodienoic acid; 13-AOS, 13-allene oxide synthase; 13-DES, 13-divinyl ether synthase; 13-HPL, 13-hydroperoxide lyase; 13-KOT, 13-fatty acid ketotriene; 13-LOX, 13-lipoxygenase; (13S)-HPOT, 13-hydroperoxy-10,12,15-octadecatrienoic acid; (13S)-HOT, 13-hydroperoxyde lyase; α-DOX, α-dioxygenases; AAT, alcohol acyltransferase; AOC, allene oxide cyclase; ADH, alcohol dehydrogenase; EAS, epoxyalcohol synthase; EAH, epoxy hydrolase; ISO, isomerase; OPR, *oxo*-phytodienoic acid reductase; POX, peroxigenase; ROS, reactive oxygen species; TriHOD, trihydroxy fatty acids.

**Table 1 cells-10-00706-t001:** Examples of lipases involved in the plant stress response *.

	Lipase	Sequence ID	Organism	Substrate
Cold Stress	PLIP2	At1g02660	*A. thaliana*	MGDG
	PGD1	Cre03.g193500	*C. reinhardtii*	MGDG
Freezing	SAG101	At5g14930	*A. thaliana*	TAG
	EDS1	At3g48090		Not determined
	PAD4	At3g52430		Not determined
Heat Stress	HIL1	At4g13550	*A. thaliana*	MGDG
Drought and Osmotic Stress	pPLAIIα	At2g26560	*A. thaliana*	Several substrates
	PLIP3	At3g62590	*A. thaliana*	PG
Pathogen Defenses	SOBER1	At4g22305	*A. thaliana*	PC / PA
	GLIP1	At5g50990		Synthetic esters
	GLIP2	At1g53940		
Oxylipin Responses	DAD1	At2g44810	*A. thaliana*	MGDG, PC, TAG
	DGL	At1g06800	*A. thaliana*	MGDG, DGDG, PC, TAG
	GLA1	FJ821553	*N. attenuata*	PC, MGDG, TAG

* References for each of the respective genes are given in the text. List of abbreviations in alphabetical order: *A. thaliana*, *Arabidopsis thaliana*; *C. reinhardtii*, *Chlamydomonas reinhardtii*; DAD1, Defective in Anther Dehiscent1; DGDG, digalactosyldiacylglycerol; DGL, Dongle; EDS1, Enhanced Disease Susceptibility1; GLA1, Glycerolipase A1; GLIP1, GDSL Lipase 1; GLIP2, GDSL Lipase 2; HIL1, Heat Inducible Lipase1; MGDG, monogalactosyldiacylglycerol; *N. attenuata*, *Nicotiana attenuata*; PAD4, Phytoalexin Deficient4; PA, phosphatidic acid; PC, phosphatidylcholine; PG, phosphatidylglycerol; PGD1, Plastid Galactoglycerolipid Degradation1; PLIP2, Plastid Lipase 2; PLIP3, Plastid Lipase 3; SAG101, Senescence-Associated Gene101; SOBER1, Suppressor of AvrBst-Elicited Resistance1; TAG, triacylglycerol.

## Data Availability

Not applicable.

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
