# Peer review of "The Role of Chloroplast Membrane Lipid Metabolism in Plant Environmental Responses"

_cells, 2021, doi:10.3390/cells10030706_

Round 1
Reviewer 1 Report
In this manuscript, the authors comprehensively summarize chloroplast membrane lipid metabolism in environmental stress responses. The manuscript is well organized and the illustrations are informative and well prepared. Please find my concerns below.
Since the authors mentioned a variety of lipases in plastids, it might be helpful for readers to understand their function if the authors can summarize these lipases in a table.
What does numbers 1-5 mean in Figure 1?
Line 489-503, a short subtitle is required.
Please rephrase the sentence in Line 585-588.
Author Response
In this manuscript, the authors comprehensively summarize chloroplast membrane lipid metabolism in environmental stress responses. The manuscript is well organized and the illustrations are informative and well prepared. Please find my concerns below.
Since the authors mentioned a variety of lipases in plastids, it might be helpful for readers to understand their function if the authors can summarize these lipases in a table.
Response: In order to facilitate the reading of the manuscript regarding the role of lipases in plant stress response, we have included in the manuscript Table 1 as suggested by reviewer 1.
What does numbers 1-5 mean in Figure 1?
Response: The numbers in Figure 1 refer to the different proteins forming the TGD complex: TGD1, TGD2, TGD3, TGD4 and TGD5. We have added this sentence in the legend of figure 1: «The numbers refer to the TGD1-5 proteins forming the TGD complex».
Line 489-503, a short subtitle is required.
We have corrected the title of section 3.1.3 («Pathogen Defenses») here.
Please rephrase the sentence in Line 585-588.
Response: Done
Reviewer 2 Report
The review focuses on the changes and role of chloroplast membrane lipids in plants during abiotic and biotic stress. The manuscript is well-written and information is presented in a comprehensive manner.
However, only phosphate stress has been mentioned in the manuscript. Does other stress impact the membrane lipids similarly ? If not, please provide information or limit the title and scope of review to phosphate stress.
Can the authors provide information on the changes in fatty acid composition ?
The authors should include a section on Future perspectives.
Author Response
The review focuses on the changes and role of chloroplast membrane lipids in plants during abiotic and biotic stress. The manuscript is well-written and information is presented in a comprehensive manner.
However, only phosphate stress has been mentioned in the manuscript. Does other stress impact the membrane lipids similarly ? If not, please provide information or limit the title and scope of review to phosphate stress.
Response: In addition to phosphate stress (2.5.), we mentioned several abiotic and biotic stresses in the initial manuscript: cold (3.1.1.) and freezing (2.6.), heat stress (3.1.1.), osmotic stress and drought (3.1.2.) and biotic stresses (3.1.3.). Table 1, suggested by reviewer 1, provides a broader overview of the stresses mentioned in this review.
Can the authors provide information on the changes in fatty acid composition ?
Response: We do not understand this comment as the entire section three makes this connection.
The authors should include a section on Future perspectives.
Round 2
Reviewer 2 Report
The authors have revised the manuscript according to the comments.